# Long-Acting FGF21 Inhibits Retinal Vascular Leakage in In Vivo and In Vitro Models

**DOI:** 10.3390/ijms21041188

**Published:** 2020-02-11

**Authors:** Yohei Tomita, Zhongjie Fu, Zhongxiao Wang, Bertan Cakir, Steve S. Cho, William Britton, Ye Sun, Ann Hellström, Saswata Talukdar, Lois E.H. Smith

**Affiliations:** 1Department of Ophthalmology, Boston Children’s Hospital, Harvard Medical School, 300 Longwood Ave, Boston, MA 02115, USA; yohei.tomita@childrens.harvard.edu (Y.T.); Zhongjie.Fu@childrens.harvard.edu (Z.F.); Zhongxiao.Wang@childrens.harvard.edu (Z.W.); Bertan.Cakir@childrens.harvard.edu (B.C.); Steve.Cho@childrens.harvard.edu (S.S.C.); William.Britton@childrens.harvard.edu (W.B.); Ye.Sun@childrens.harvard.edu (Y.S.); 2Manton Center for Orphan Disease, Boston Children’s Hospital, 300 Longwood Ave, Boston, MA 02115, USA; 3Section for Ophthalmology, Department of Clinical Neuroscience, Institute of Neuroscience and Physiology, Sahlgrenska Academy, University of Gothenburg, Box 100, SE-405 30 Gothenburg, Sweden; ann.hellstrom@medfak.gu.se; 4Merck & Co., Inc., 213 East Grand Avenue, South San Francisco, CA 94080, USA; saswata.talukdar@merck.com

**Keywords:** fibroblast growth factor 21 (FGF21), tight junction, Claudin-1, vascular endothelial growth factor (VEGF), vascular leakage, machine learning analysis

## Abstract

The aim of the current study was to investigate the impact of long-acting fibroblast growth factor 21 (FGF21) on retinal vascular leakage utilizing machine learning and to clarify the mechanism underlying the protection. To assess the effect on retinal vascular leakage, C57BL/6J mice were pre-treated with long-acting FGF21 analog or vehicle (Phosphate Buffered Saline; PBS) intraperitoneally (i.p.) before induction of retinal vascular leakage with intravitreal injection of mouse (m) vascular endothelial growth factor 164 (VEGF164) or PBS control. Five hours after mVEGF164 injection, we retro-orbitally injected Fluorescein isothiocyanate (FITC) -dextran and quantified fluorescence intensity as a readout of vascular leakage, using the Image Analysis Module with a machine learning algorithm. In FGF21- or vehicle-treated primary human retinal microvascular endothelial cells (HRMECs), cell permeability was induced with human (h) VEGF165 and evaluated using FITC-dextran and trans-endothelial electrical resistance (TEER). Western blots for tight junction markers were performed. Retinal vascular leakage in vivo was reduced in the FGF21 versus vehicle- treated mice. In HRMECs in vitro, FGF21 versus vehicle prevented hVEGF-induced increase in cell permeability, identified with FITC-dextran. FGF21 significantly preserved TEER compared to hVEGF. Taken together, FGF21 regulates permeability through tight junctions; in particular, FGF21 increases Claudin-1 protein levels in hVEGF-induced HRMECs. Long-acting FGF21 may help reduce retinal vascular leakage in retinal disorders and machine learning assessment can help to standardize vascular leakage quantification.

## 1. Introduction

Diabetic retinopathy (DR) is a major cause of vision impairment with increasing prevalence worldwide [1]. Diabetic macular edema (DME), which occurs secondary to retinal vascular leakage, is the leading cause of vision loss in DR. Vascular endothelial growth factor (VEGF) plays a key role in DME [2]. Vitreous VEGF concentration is higher in patients with DME compared to non-DR patients [3]. Anti-VEGF and steroid therapies are widely used to treat DME. Although these therapeutic approaches have improved the visual prognosis of DME, many cases are non-responders. In addition, these treatments carry risks, such as cataract formation, glaucoma, and endophthalmitis [4,5,6]. Therefore, it is important to develop alternative therapeutic approaches to treat DME, optimally with an intervention that suppresses both neovascularization and vascular leakage.

Fenofibrate (which has been shown to effectively suppress progression of DR) may act through fibroblast growth factor (FGF21). Two large randomized controlled trials—the Fenofibrate Intervention and Event Lowering in Diabetes (FIELD) study and the Action to Control Cardiovascular Risk in Diabetes (ACCORD) study—revealed that a fibric acid derivative, fenofibrate, inhibits the progression of DR and other microvascular endpoints in patients with Type 2 diabetes [7,8]. Fenofibrate is primarily used clinically as a hypolipidemic agent that substantially reduces triglyceride levels and increases high-density lipoprotein (HDL)-cholesterol levels by activating the peroxisome proliferator-activated receptor alpha (PPARα). Although activation of PPARα by fenofibrate suppresses pathological retinal angiogenesis in rodent models, precise mechanisms of fenofibrate activation of PPARα in this process are not yet fully understood [9,10].

Recently, several groups have shown that administration of a PPARα agonist increases fibroblast growth factor 21 (FGF21) [11,12]. A recent study shows that a selective PPARα modulator (SPPARMα) prevents pathological neovascularization in a mouse model by increasing circulating FGF21 [12]. FGF21 is a secreted protein that is comprised of 209 amino acids [13]. FGF21 reduces body weight and improves the lipid profile in patients with Type 2 diabetes, as well as in primate and rodent models of Type 2 diabetes [14,15]. We have reported that long-acting FGF21, with a longer half-life than native FGF21, prevents pathological neovascularization in the retina and choroid in several murine models, and also protects photoreceptor function in diabetic mice [16,17]. However, the impact of FGF21 on retinal vascular leakage has not yet been explored.

Mouse models of human DME are limited in large part because mice have no macula. In mouse models of DR or of retinal neovascularization (oxygen-induced retinopathy), retinal vascular leakage is minimal compared to that of humans [18]. Therefore, retinal vascular leakage is commonly induced by intravitreally injecting VEGF or utilizing induced VEGF endogenous expression, and leakage visualized with Fluorescein isothiocyanate (FITC) dextran perfusion, and quantified visually by individual readers [19,20]. In addition, the quantification process with Evans Blue is variable in mice [21,22]. Thus, we used an unbiased machine learning program to reliably characterize retinal vascular leakage in mice. Recently, a study showed the utility of machine learning and deep learning neural networks for quantification of retinal neovascularization and vaso-obliteration in oxygen induced retinopathy in mice. Machine learning and deep learning methods could improve efficiency and objectivity compared to conventional whole-mount grading systems [23]. This study utilized this machine learning system to investigate the impact of long-acting FGF21 on retinal vascular leakage and to clarify the mechanism underlying the protection.

## 2. Results

### 2.1. Establishing a Machine Learning Program for Vascular Leakage Quantification

FITC-dextran-perfused retinal whole-mounts were isolated and visualized using confocal microscopy. The machine learning program was trained over multiple trials such that every object in an image was individually categorized with a different color. The model then performed image segmentation as shown (Figure 1A–C). For image categorization, we used red for vessels, yellow for leakage, and cyan for background. Whole-mounted retinas were imaged with confocal microscopy and segmented with the machine learning software, as shown in (Figure 1D,E).

### 2.2. FGF21 Agonist Inhibits and FGF21 Deficiency Increases VEGF-Induced Retinal Vascular Leakage in Mice

C57BL/6J mice were divided into three treatment groups: intravitreal vehicle (Phosphate Buffered Saline; PBS) + intraperitoneal (i.p.) vehicle (PBS), intravitreal mouse (m) VEGF + i.p. vehicle (PBS), and intravitreal mVEGF + i.p. FGF21 (long-acting FGF21; PF05231023; Figure 2A). The FITC-dextran-stained retinal whole-mounts were first isolated and imaged (Figure 2B–D). Retinal vascular leakage was quantified using the machine learning system (Figure 2E–G). We then compared the intensity of the retinal leakage among the three groups. We found that leakage was induced in mice that received an intravitreal injection of mVEGF versus vehicle control (*p* = 0.046; *n* = 9,14). FGF21 (i.p.) versus vehicle (PBS) pre-treatment reduced mVEGF-induced retinal vascular leakage (*p* = 0.01; *n* = 14,15; Figure 2H). We examined the impact of FGF21 deficiency on leakage by comparing intravitreal mVEGF-treated wild-type (*Fgf21^+/+^*) and knockout (*Fgf21^−/−^*) littermate mice (Figure 2I–L). *Fgf21^−/−^* mice had increased permeability compared to Fgf21^+/+^ mice (Figure 2M).

### 2.3. FGF21 Decreases HRMECs Permeability in Vitro

We found that receptors for FGF21 were expressed in human (h) primary retinal microvascular endothelial cells (HRMECs) in vitro: Fibroblast growth factor receptor 1 (FGFR1) at a high level, FGFR3 and co-receptor β-KLOTHO (validated with Huh7, a human hepatocellular carcinoma cell line as a positive control and water as a negative control) at moderate levels, and FGFR2 and FGFR4 at low levels (Appendix A). Human (h) VEGF treatment (10 ng/mL) in HRMECs increased permeability (identified with FITC-dextran) in cells compared with control treatment (*p* = 0.0006). FGF21 (50 ng/mL, 100 ng/mL) versus vehicle (PBS) pre-treatment prevented hVEGF-induced increases in permeability (*p* = 0.001; Figure 3A–C).

### 2.4. FGF21 Stabilizes TEER in HRMECs in Vitro

We further assessed HRMECs barrier function with trans-endothelial electrical resistance (TEER) measurements. Initially we determined the TEER time-dependency after hVEGF treatment. The TEER was reduced eighteen hours after treatment with hVEGF (10 ng/mL) versus vehicle control (*p* = 0.005; Figure 3D,E). To examine the impact of FGF21 on HRMECs TEER, we pre-treated the cells with FGF21 or vehicle (PBS) five hours before hVEGF treatment. We checked TEER eighteen hours after adding hVEGF (Figure 3F). FGF21 (100 ng/mL) versus vehicle control preserved TEER in hVEGF-induced HRMECs (*p* = 0.002; Figure 3G).

### 2.5. FGF21 Preserves Expression of Tight Junction (Claudin-1) Protein in HRMECs

To evaluate the role of tight junctions in the protective effect of FGF21 against retinal vascular leakage, we examined tight and adhesion junction (Claudin-1,5, VE-cadherin, Occludin) protein levels in hVEGF-treated HRMECs. hVEGF treatment decreased Claudin-1 protein levels (*p* = 0.048), while FGF21 preserved Claudin-1 levels following hVEGF treatment in HRMECs (*p* = 0.035; *n* = 3; Figure 4A). FGF21 did not change the levels of Claudin-5, VE-cadherin, and Occludin in hVEGF-treated HRMECs (Figure 4B–D; *n* = 3–7). hVEGF treatment decreased *CLDN-1* mRNA levels (*p* < 0.05), while FGF21 tended to preserve *CLDN-1* mRNA levels following hVEGF treatment (*p* = 0.06) (Appendix A). FGF21 did not change the mRNA levels of *CLDN-5*, *VE-CADHERIN*, and *OCLN* in hVEGF-treated HRMECs (Appendix A).

## 3. Discussion

In this study, we found that long-acting FGF21 reduced vascular permeability in hVEGF-treated HRMECs. FGF21 restored tight junction protein Claudin-1 levels which were reduced with hVEGF treatment (Figure 5). We also found that in vivo, long-acting FGF21 protected against mVEGF-induced retinal vascular leakage, as analyzed objectively by a machine learning system.

The traditional leakage assay in mice with human assessment of degree of leakage has high variability. We found that the machine learning assay, based on a program provided by Zeiss, could determine the effect of FGF21 in suppressing vascular leakage. To train the software, only a few images were needed because the algorithm is based on pixel-based classification, not on object-based classification. Pixel-based models are generally easier to train than object-based classification models which need more initial training processes. Based on the initial labeling, a neural net is trained automatically, and this model can be used repeatedly to segment different datasets. In addition, this method minimizes errors that may result from user bias.

In vivo, mVEGF intravitreal treatment increased retinal vascular leakage which was suppressed with systemic (i.p.) FGF21. In addition, *Fgf21^−/−^* retinas showed increased retinal vascular permeability/leakage compared to *Fgf21^+/+^* retinas. Together, these data indicate that FGF21 treatment suppresses mVEGF-induced retinal vascular leakage in this model. In an in vitro endothelial cell permeability assay, hVEGF treatment increased permeability compared to control and long-acting FGF21 treatment decreased permeability.

Permeability is dependent on the transport of molecules through the blood-retinal barrier which can occur via two pathways: the transcellular route which consists of transporters; and the paracellular route, which consists of tight junctions, adherence junctions, and gap junctions [24]. To identify whether FGF21 regulates permeability through tight junctions, we measured TEER, to assess barrier function between cells, employing the electrical resistance measurements. Given that tight junction proteins influence resistance between the apex and the basal side of the cell, we co-treated the endothelial cells with hVEGF and FGF21/vehicle to calculate any change in the resistance. We found that hVEGF decreased TEER while 100 ng/mL FGF21 preserved TEER in HRMECs. These results suggest that FGF21 regulates permeability through tight junctions.

To investigate the tight junction proteins possibly involved in the permeability effect of FGF21, we quantified levels of VE-Cadherin, Occludin, and Claudin-1 and 5, and found FGF21 effects only on Claudin-1, which is one of the tight junction proteins in the Claudin family [25]. Claudin-1 is present in the tight junctions between endothelial cells of retinal vessels [26,27]. In addition, Claudin-1 gene expression is suppressed following VEGF treatment [28]. Our findings showed that hVEGF treatment decreased Claudin-1 levels while FGF21 co-treatment preserved Claudin-1 levels. We did not find any changes in Claudin-5 in hVEGF-treated HRMECs. This result is consistent with the report that Claudin-1, but not Claudin-5, is lost after VEGF treatment in HRMECs [26].

In our in vitro endothelial cell study, we detected *FGFR1* and *β -KLOTHO,* which are crucial receptors for FGF21 functionality (Appendix A) [29,30]. We also detected moderate expression of *FGFR3*, and very low expression of *FGFR2* and *FGFR4* (Appendix A). In our in vivo study, there may be potential systemic FGF21 effects on vascular leakage. We reported earlier that *Fgfr1-4* and *β-Klotho* are expressed in the total retina and present in the retinal vessels [16]. Thus, we speculate that FGF21 may have a therapeutic effect on the retina through a direct effect on vascular endothelial cells.

Although our study applied an unbiased method of leakage quantification and identified a possible pathway in which FGF21 protects against vascular leakage, it has several limitations. The VEGF-induced retinal vascular leakage assay does not model non-VEGF-induced leakage. However, other methods such as the lipopolysaccharide (LPS) injection model, which induces leakage after severe inflammation, is variable, and the induced extreme inflammation does not reflect the cause of leakage seen in diabetic retinopathy appropriately. In the Akimba mouse, another diabetic animal model with VEGF expression in the eye, leakage from retinal vessels is also based on VEGF, thus offering no clear advantage over the VEGF injection model [31,32].

Furthermore, the mechanism behind FGF21 regulation of vascular leakage may not depend solely on Claudin-1. We focused on endothelial cells in our current study, but vascular leakage may also be mediated via other cells such as pericytes, microglia, Muller glial cells, and astrocytes, which are components of the neurovascular unit [33]. A previous study showed that FGFR1 is found in ganglion cells and Müller cells in the mouse retina [34]. In addition, FGFR-1 was strongly labeled in all nuclear layers in the human retina [35]. These results suggest the inner neural cells may also be involved in the mechanism. Further studies are needed to define the role of these other cells in vascular leakage.

Our machine learning approach provides a fast, unbiased, reproducible, and quantitative way to examine vascular leakage. This method can also be adapted and applied in many other fields with retinal area quantification based on the fluorescent intensity. For example, machine learning and deep learning can overcome individual biases of quantifying avascular and neovascular area in the oxygen-induced retinopathy model, as well as mitigate time and labor using Image J and Photoshop [23].

In addition to our current finding that FGF21 inhibits vascular leakage, we have previously reported that FGF21 also inhibits pathological choroidal and retinal neovascularization [16], as well as protects against neuronal degeneration in T1D mice [17]. Taken together, we propose that FGF21 could be a therapeutic target for preventing macular edema and proliferative retinopathies.

## 4. Materials and Methods

### 4.1. Ethics Statement

C57BL/6J mice were purchased from the Jackson Laboratory. *Fgf21^+/+^* and *Fgf21^−/−^* mice were provided by Drs. Steven Kliewer and David Mangelsdorf from the University of Texas Southwestern. Both male and female mice were used. All animal studies adhered to the Association for Research in Vision and Ophthalmology Statement for the Use of Animals in Ophthalmic and Vision Research and were approved by Institutional Animal Care and Use Committee (IACUC) at Boston Children’s Hospital (protocol number 19-04-3913R, approved on 2 December 2019).

### 4.2. Animal Model

Six to eight-week-old male C57BL/6J mice were pre-treated with a long-acting FGF21 analog PF-05231023 (10 mg/kg, #530533, MedKoo Biosciences, Morrisville, NC, USA) or vehicle (Phosphate-Buffered Saline; PBS) via intraperitoneal (i.p.) injection twice a week for one week. Retinal vascular leakage was then induced by intravitreal administration of mVEGF164 (493-MV, R&D system, Minneapolis, MN, USA, 100 ng/ul, 1µl). The control group was injected i.p. with PBS twice a week for a week, then intravitreal PBS (control) was injected. Six to eight-week-old male and female *Fgf21^+/+^* and *Fgf21^−/−^* mice were also evaluated. Five hours after the mVEGF or PBS injection, we retro-orbitally delivered FITC-dextran (70kDa, 100 mg/mL, Cat. # D1822, Thermo Fisher Scientific, Waltham, MA, USA). One hour after the FITC-dextran injection, eyes were enucleated, and the retinas were whole-mounted (Figure 2A).

### 4.3. Imaging and Machine Learning Analysis

The freshly dissected and whole-mounted retinas were imaged on a confocal microscope (LSM880, Zeiss, Oberkochen, Germany) and Z-stack scanning was performed (deep, intermediate, and superficial layer). The fluorescence intensity of FITC-dextran from vascular leakage was analyzed with the machine learning software “ZEISS Efficient Navigation (ZEN) Intellesis”, to segment and quantify leakage. In brief, the segmentation module ZEN Intellesis was trained to use machine learning to automatically identify objects within an image, according to a predefined set of the model. This enables microscope users to perform image segmentation—even on complex data sets—without any programming experience or advanced knowledge of how to set up the image segmentation. The result was then integrated into the ZEN image analysis workflow, and data were extracted. We labeled objects and trained the model several times with multiple images, until it was able to perform image segmentation of the whole retina and derive the intensity from each segmentation (Figure 1). We then compared the intensity of the leakage among three groups: intravitreal vehicle (PBS) + i.p. vehicle (PBS), intravitreal mVEGFA + i.p. PBS, and intravitreal mVEGFA + i.p. FGF21 treatment.

### 4.4. Endothelial Monolayer Permeability Assay

Primary human retinal microvascular endothelial cells (HRMECs, ACBRI181, Cell Systems, Kirkland, WA, USA) were cultured in Endothelial Cell Growth Medium MV2 (C-22022, PromoCell GmbH, Heidelberg, Germany). The cells were seeded on the inner surface of collagen-coated Transwell inserts (6.5 mm diameter, 0.4 μm pore size polycarbonate filter; Corning, Corning, NY, USA), which were then placed in wells of a 24-well plate with complete MV2 media. The culture medium was changed every other day. After the cells reached to confluence and polarized in the upper chamber, FGF21 (or PBS) was added to the upper and lower chambers. Five hours later, human (h) VEGF165 (10 ng/mL, 293-VE, R&D system, Minneapolis, MN, USA) was added to both chambers, to increase cell permeability. FITC-dextran (70kDa, 0.5 mg/mL) was added to the upper chamber at eighteen hours after hVEGF treatment. One hour after adding FITC-dextran, media were collected from the bottom chambers and the fluorescence intensity of FITC-dextran was measured (at 538 nm; Figure 3A).

### 4.5. Trans-Endothelial Electrical Resistance Assay

Trans-endothelial electrical resistance (TEER) assesses barrier function between cells and was measured using an Electrical Resistance System (Millicell ERS-2, MilliporeSigma, Burlington, MA, USA) [36], as per manufacturer’s instructions. First, we checked the TEER to determine a time dependency after induction with hVEGF. Then we checked TEER 18 hours after adding hVEGF (Figure 3F). Cultures were stabilized at room temperature for 15 min before TEER measurement. The TEER of the monolayer was calculated as follows:

TEER of the monolayer (ΔΩcm^2^) = (sample-well resistance-blank-well resistance) (Ω) × area of the cell monolayer (cm^2^)

Each group contained triplicate cultures, and each experiment was repeated at least twice, using different batches of cells.

### 4.6. Western Blotting

Protein was extracted from HRMECs treated with a vehicle control (1% PBS), hVEGF (10 ng/mL) or hVEGF and FGF21 (100 ng/mL) in radioimmunoprecipitation assay buffer (RIPA; #89900; Pierce, Grand Island, NY, USA) supplemented with protease inhibitor (1:100, P8340, Sigma-Aldrich, St. Louis, MO, USA); an equal amount of protein was separated in a 4–12% gel and transferred to nitrocellulose membranes. The membranes were incubated overnight with primary anti-Claudin-1 antibody (1:1000; #71-7800, Thermo Fisher Scientific, Waltham, MA, USA), anti-Clauin-5 (1:1000, #35-2500, Thermo Fisher Scientific, Waltham, MA, USA), Occludin (1:1000; ab216327, Abcam, Cambridge, UK), and VE-cadherin (1:1000, ENZ-ABS661, Enzo Life Sciences, Inc., Farmingdale, NY, USA). Signals were detected using corresponding horseradish peroxidase-conjugated secondary antibodies (1:5000, NA934V, NA9310V, GE Healthcare Ltd., Great North Road Hatfield, UK) and enhanced chemiluminescence (#34095, Thermo Fisher Scientific, Waltham, MA, USA). GAPDH (1:1000, sc-32233; Santa Cruz, Dallas, TX, USA) was used as an internal control.

### 4.7. Real-Time PCR

Real-time PCR was performed as a standard protocol. HRMECs were lysed with QIAzol lysis reagent and incubated on ice for 15 min. Chloroform 20% was added and incubated for 5 min at room temperature. The mixture was centrifuged at 12,000 g for 15 min, and the supernatant was collected for RNA extraction according to the manufacturer’s instructions using a PureLink RNA Mini Kit (#12183018A; Invitrogen, Grand Island, NY, USA). RNA was then reverse transcribed using iScript cDNA synthesis kit (#1708891; BioRad, Hercules, CA, USA). Gene expression (mRNA) was quantified using an Applied Biosystems 7300 Real-Time PCR system (Thermo Fisher Scientific, Waltham, MA, USA) with the SYBR Green Master mix kit (bimake.com, Houston, TX, USA). Gene expression was calculated relative to 18S and *CYCLO-A* and *β-ACTIN* internal control using the ΔΔCt method. The relative mRNA levels were presented as the ratio of change versus internal control. Oligonucleotide primers are listed in Appendix A.

### 4.8. Statistical Analysis

Animal data are presented as mean ± standard error (SE). Two-tailed unpaired *t*-test, and ANOVA with Tukey’s multiple comparison test were used to compare the results as specified in the figure legends (Prism v8.0; GraphPad Software, Inc., San Diego, CA, USA). *p* < 0.05 is considered as statistically significant.

## 5. Conclusions

In summary, we have shown, using a machine learning algorithm, that FGF21 may help prevent VEGF-induced retinal vascular leakage by stabilizing vascular endothelial cell tight junctions by increasing Claudin-1. Long-acting FGF21 may have therapeutic potential for preventing or treating macular edema in retinopathy as well as suppressing retinal neovascularization [16]. Furthermore, we hope that the machine learning assessment described in this study can help to better standardize vascular leakage quantification.

## Figures and Tables

**Figure 1 ijms-21-01188-f001:**
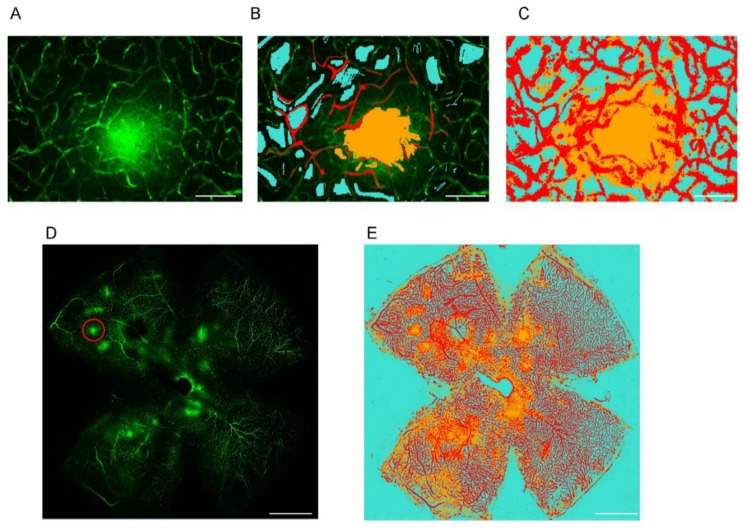
Representative images of whole mounted retina after labeling with machine learning. (**A**) Magnification of the area within the red circle in (**D**). (**B**) Labeling of each segment. (**C**) After analyzing the image. Scale bar is 100 µm. (**D**) Original image of retina using confocal microscope. (**E**) Image of retina after labeling. The yellow shows leakage, the red shows vessels and artifacts, and the cyan shows background. Scale bar is 1 mm.

**Figure 2 ijms-21-01188-f002:**
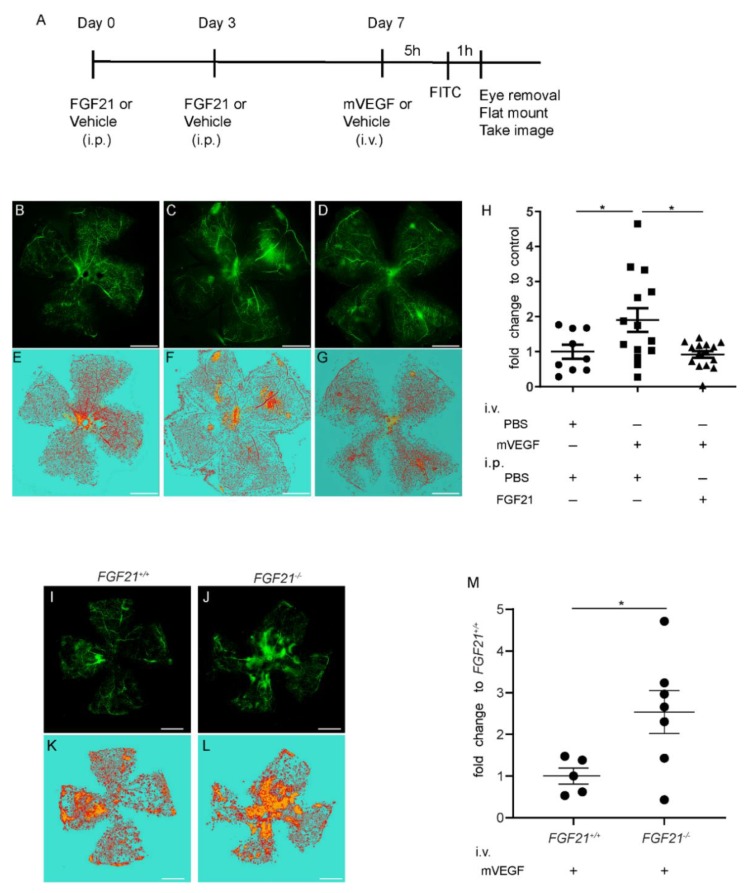
Fibroblast growth factor 21 **(**FGF21) treatment prevents and FGF21 deficiency Increases vascular endothelial growth factor (VEGF) -induced retinal vascular leakage in the mouse model. (**A**) The time course of this study. (**B**–**D**) Representative confocal microscopy images of whole-mounted retina. (**E**–**G**) Images of retina after segmentation with ZEISS Efficient Navigation (ZEN) Intellesis. (**B** and **E**) Representative retinal image of controls. (**C** and **F**) Control Phosphate Buffered Saline (PBS) treatment with intravitreal injection of mouse (m) VEGF. (**D** and **G**) FGF21 (long-acting FGF21; PF05231023) treatment with intravitreal injection of mouse (m) VEGF. The yellow FITC indicates leakage; the red lectin stain indicates vessels and artifacts; the cyan shows background. (**H**) The intensity of retinal vascular leakage was increased in animals injected with mVEGF and treated with PBS (square), compared to controls (circle, *n* = 9–14; *p* = 0.046). Leakage intensity was reduced in FGF21 (triangle) versus PBS-treated groups (*n* = 14–15; *p* = 0.01). The data were analyzed by one-way ANOVA with Tukey and were expressed as mean ± standard error (SE) Scale bar is 1 mm. (**I** and **J**) Representative confocal microscopy images of retinas in *Fgf21^+/+^* and *Fgf21^−/−^* littermate mice. Representative images are shown. (**K** and **L**) Images of retina after segmentation with ZEN Intellesis. (**M**) The data were analyzed by unpaired *t*-test and were expressed as mean ± SE. Fold change compared with the wild type group was calculated (*n* = 5–7; *p* = 0.037). Scale bar is 1mm. i.p., intraperitoneal injection, i.v., intravitreal injection, FGF21, long-acting FGF21 (PF05231023). Fold change was calculated (* *p* < 0.05).

**Figure 3 ijms-21-01188-f003:**
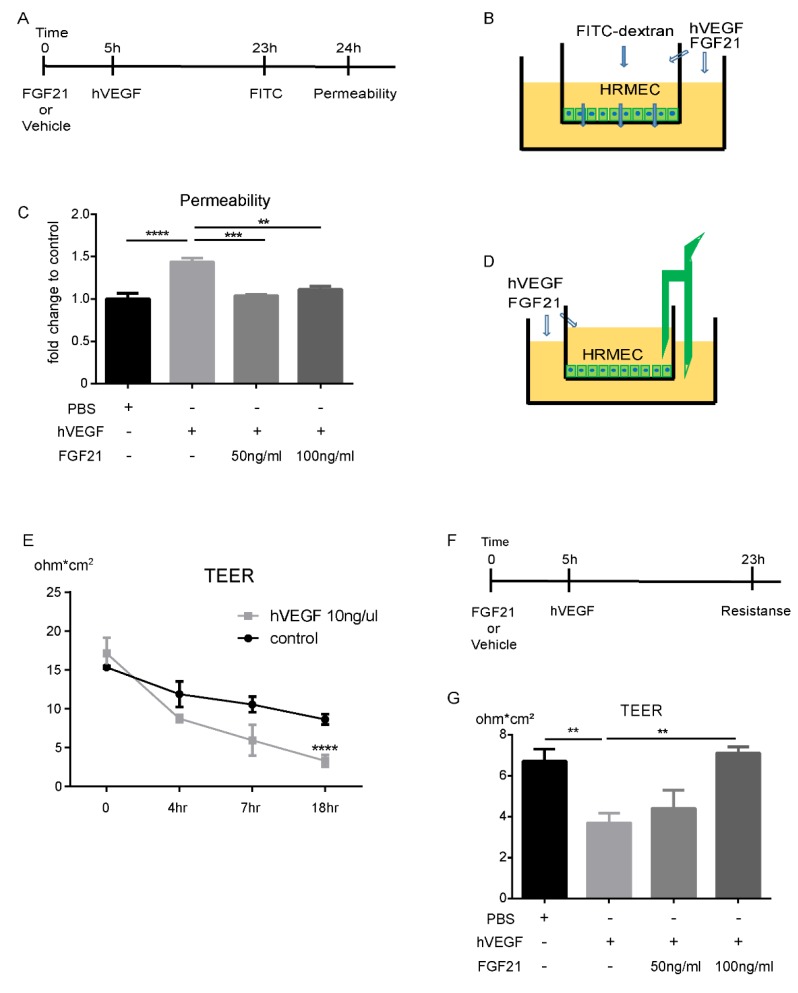
FGF21 decreases vascular leakage in primary human retinal microvascular endothelial cells (HRMECs). (**A**) The time course of permeability assay. (**B**) Experimental schema of this assay. (**C**) human (h)VEGF increased permeability in HRMECs, compared with controls (*n* = 5, *p* < 0.0001). Long-acting FGF21 (50 ng/mL, 100 ng/mL) prevented hVEGF-induced increase in cell permeability, identified with Fluorescein isothiocyanate (FITC) -dextran (*n* = 3–5, *p* = 0.0006, 0.001, respectively). (**D**) The schema of trans-endothelial electrical resistance (TEER) experiment. (**E**) hVEGF reduced TEER eighteen hours after administration, compared with controls (*p* < 0.001). (**F**) The time course of TEER experiment. (**G**) hVEGF (10 ng/mL) decreased TEER compared to control (*p* = 0.005). FGF21 (100 ng/mL) preserved TEER compared to hVEGF + PBS treatment (*p* = 0.002). The data were analyzed by one-way ANOVA with Tukey and were expressed as mean ± SE (** *p* < 0.01, *** *p* < 0.001, **** *p* < 0.0001).

**Figure 4 ijms-21-01188-f004:**
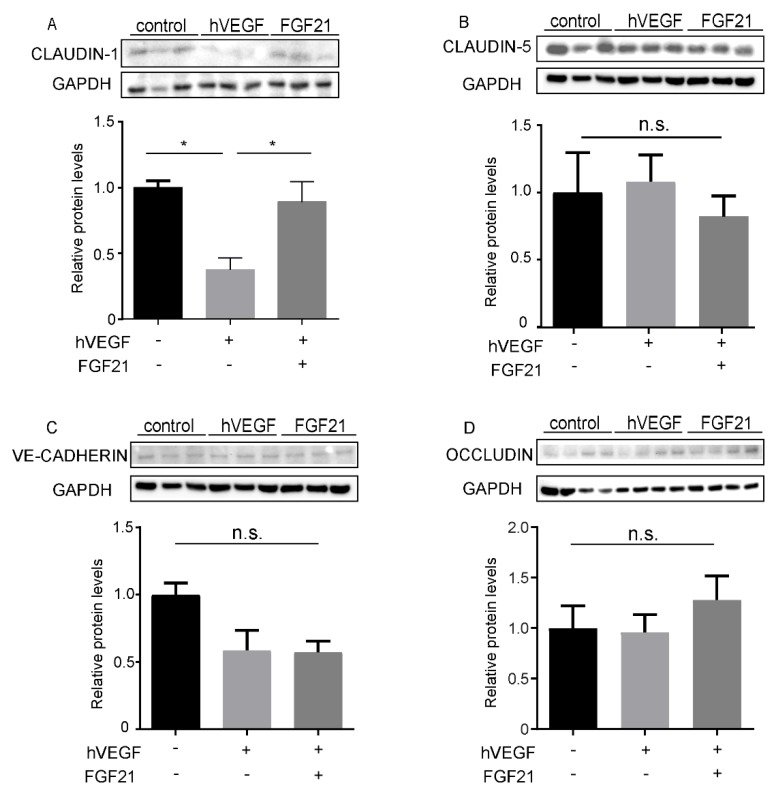
FGF21 preserves Claudin-1 expression in HRMECs. (**A**) A representative image of Claudin-1 Western blotting. hVEGF treatment decreased levels of Claudin-1 protein (*p* = 0.048), while FGF21 preserved Claudin-1 protein expression levels following hVEGF treatment (*p* = 0.035). (**B**) A representative image of Claudin-5 Western blotting. There was no significant difference among the three groups. (**C**) A representative image of VE-cadherin Western blotting. There was no significant difference among the three groups. (**D**) A representative image of Occludin Western blotting. There was no significant difference among the three groups. Data were analyzed by one-way ANOVA with Tukey and were expressed as mean ± SE. n.s., not significant. Fold change was calculated (* *p* < 0.05).

**Figure 5 ijms-21-01188-f005:**
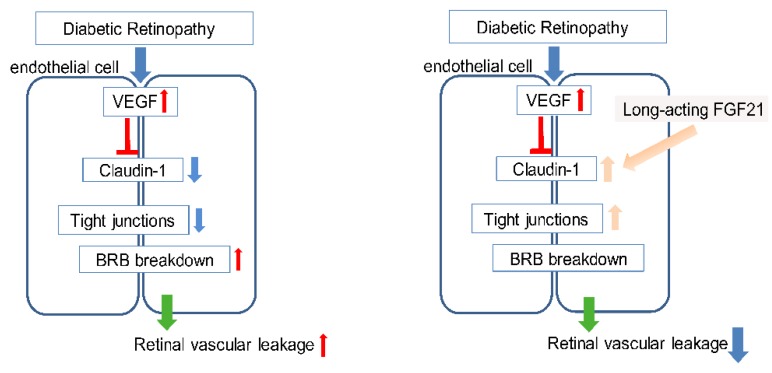
Schematic of the proposed pathway. VEGF suppresses Claudin-1 and disrupts the blood-retinal barrier (BRB) with increased vascular leakage in the retina. FGF21 may prevent VEGF-induced retinal vascular leakage by stabilizing tight junctions via Claudin-1.

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
