# Peer review of "Long-Acting FGF21 Inhibits Retinal Vascular Leakage in In Vivo and In Vitro Models"

_ijms, 2020, doi:10.3390/ijms21041188_

Round 1

Reviewer 1 Report

1.Authors report that “Anti-VEGF and steroid therapies, which are widely used to treat DME, are often ineffective.”. These sentence should be changed underlining how even if the actual therapeutic approaches have changed positively the visual prognosis of the patients with DME, to date many cases are no responders, or could show some side effects or complications related to the way the drug is administered.

2.Authors should add the main limits or possible bias of the current study in the conclusion section.

3.In the discussion section, authors should explain more in details the possible future applications of the machine learning algorithm they used, together with the long-acting FGF21 therapeutic potential use for preventing or treating macular edema in retinopathy as well as suppressing retinal neovascularization.

Author Response

Response to Reviewer 1 Comments

1.Authors report that “Anti-VEGF and steroid therapies, which are widely used to treat DME, are often ineffective.”. These sentences should be changed underlining how even if the actual therapeutic approaches have changed positively the visual prognosis of the patients with DME, to date many cases are no responders, or could show some side effects or complications related to the way the drug is administered.

Thank you very much for the suggestion. We have changed the sentence below. Page 1, Line 39-40

1. Introduction

Diabetic retinopathy (DR) is a major cause of vision impairment with increasing prevalence worldwide. Diabetic macular edema (DME), which occurs secondary to retinal vascular leakage, is the leading cause of vision loss in DR. Vascular endothelial growth factor (VEGF) plays a key role in DME. Vitreous VEGF concentration is higher in patients with DME compared to non-DR patients. Anti-VEGF and steroid therapies, which are widely used to treat DME. Although these therapeutic approaches have improved the visual prognosis of DME, many cases are non-responders. In addition, these treatments carry risks, such as cataract formation, glaucoma, and endophthalmitis. Therefore, it is important to develop alternative therapeutic approaches to treat DME, optimally with an intervention that suppresses both neovascularization and vascular leakage.

2.Authors should add the main limits or possible bias of the current study in the conclusion section.

Thank you for the important comment. We have added limitations of the study in the discussion section. Page 7-8, Line 222-237

3. Discussion

Although our study applied an unbiased method of leakage quantification and identified a possible pathway in which FGF21 protects against vascular leakage, it has several limitations. The VEGF-induced retinal vascular leakage assay does not model non-VEGF-induced leakage. However, other methods such as the LPS injection model, which induces leakage after severe inflammation, is variable, and the induced extreme inflammation does not reflect the cause of leakage seen in diabetic retinopathy appropriately. In the Akimba mouse, another diabetic animal model with VEGF expression in the eye, leakage from retinal vessels is also based on VEGF, thus offering no clear advantage over the VEGF injection model [31,32].

Furthermore, the mechanism behind FGF21 regulation of vascular leakage may not depend solely on Claudin-1. We focused on endothelial cells in our current study, but vascular leakage may also be mediated via other cells such as pericytes, microglia, Muller glial cells, and astrocytes, which are components of the neurovascular unit [33]. A previous study showed that FGFR1 is found in ganglion cells and Müller cells in the mouse retina [34]. In addition, FGFR-1 was strongly labeled in all nuclear layers in the human retina [35]. These results suggest the inner neural cells may also be involved in the mechanism. Further studies are needed to define the role of these other cells in vascular leakage.

3.In the discussion section, authors should explain more in details the possible future applications of the machine learning algorithm they used, together with the long-acting FGF21 therapeutic potential use for preventing or treating macular edema in retinopathy as well as suppressing retinal neovascularization.

Thank you very much for your kind suggestion. We have added the sentences below in the discussion.

Page 8, Line 238-247

3. Discussion

Our machine learning approach provides a fast, unbiased, reproducible and quantitative way to examine vascular leakage. This method can also be adapted and applied in many other fields with retinal area quantification based on the fluorescent intensity. For example, machine learning and deep learning can overcome individual biases of quantifying avascular and neovascular area in the oxygen-induced retinopathy model, as well as mitigate time and labor using Image J and Photoshop [23].

In addition to our current finding that FGF21 inhibits vascular leakage, we have previously reported that FGF21 also inhibits pathological choroidal and retinal neovascularization [16], as well as protects against neuronal degeneration in T1D mice [17]. Taken together, we proposed that FGF21 could be a therapeutic target for preventing macular edema and proliferative retinopathies.  

Reviewer 2 Report

The authors report that long-acting FGF21 inhibits retinal vascular leakage in mouse and in human retinal microvascular endothelial cells (HRMECs). This manuscript includes some interesting results; however, I have several concerns for the authors to address.

Major points:

1. The results clearly show that FGF21 suppresses VEGF-induced retinal vascular leakage. However, under diabetic conditions, multiple mechanisms including VEGF contribute to the blood-retinal barrier breakdown. Therefore, it would be important to investigate the effects of long-acting FGF21 on diabetic retinas.

2. In HRMECs, VEGF decreases Claudin-1 levels and FGF21 suppress this. Is this phenomenon observed under in vivo conditions?

3. Please discuss the mechanisms by which FGF21 suppresses VEGF-induced deceases in Claudin-1 levels.

4. In in vivo model, long-acting FGF21 is administered systemically. Please exclude the involvement of side effects on peripheral organs by systemic administration of FGF21.

5. Did the authors check that long-acting FGF21 can suppress the increased vascular permeability in Fgf21-/- mice?

6. The authors describe “We reported earlier that Fgfr1-4 and b-Klotho are expressed in the total retina and present in the retinal vessels [16]. (Page7)”. However, FGFR 1, a FGF21 receptor, has been detected in inner retina of the mouse and rat. Please discuss the inner neural cells-mediated effects.

References:Kinkl N, Hageman GS, Sahel JA, Hicks D. Fibroblast growth factor receptor (FGFR) and candidate signaling molecule distribution within rat and human retina. Mol Vis. 2002, 14,149-60.

Catalani E, Tomassini S, Dal Monte M, Bosco L, Casini G. Localization patterns of fibroblast growth factor 1 and its receptors FGFR1 and FGFR2 in postnatal mouse retina. Cell Tissue Res. 2009, 336, 423-38.

Minor points:

1.Please include the information on the age of the Fgf21+/+ and Fgf21-/- mice used in this study.

2. 4.6. Western blotting; Please include the information on incubation time and concentrations of hVEGF and FGF21.

3. Figure 3; In panels C and G, “mVEGF” should be “hVEGF”.

4. Figure 3; In panel G, “2” in cm2. Apply superscript.

5. The upper gel image of Figure 4A; the quality of gel image may be reduced by overexposure or extensive manipulation. Please replace.
